# Antimicrobial Stewardship Using Biomarkers: Accumulating Evidence for the Critically Ill

**DOI:** 10.3390/antibiotics11030367

**Published:** 2022-03-09

**Authors:** Evdoxia Kyriazopoulou, Evangelos J. Giamarellos-Bourboulis

**Affiliations:** 12nd Department of Critical Care Medicine, National and Kapodistrian University of Athens, 12462 Athens, Greece; ekyri@med.uoa.gr; 24th Department of Internal Medicine, National and Kapodistrian University of Athens, 12462 Athens, Greece

**Keywords:** antimicrobial stewardship, sepsis, COVID-19, ICU, procalcitonin, C-reactive protein, presepsin, infection, biomarker, guided antimicrobial therapy

## Abstract

This review aims to summarize current progress in the management of critically ill, using biomarkers as guidance for antimicrobial treatment with a focus on antimicrobial stewardship. Accumulated evidence from randomized clinical trials (RCTs) and observational studies in adults for the biomarker-guided antimicrobial treatment of critically ill (mainly sepsis and COVID-19 patients) has been extensively searched and is provided. Procalcitonin (PCT) is the best studied biomarker; in the majority of randomized clinical trials an algorithm of discontinuation of antibiotics with decreasing PCT over serial measurements has been proven safe and effective to reduce length of antimicrobial treatment, antibiotic-associated adverse events and long-term infectious complications like infections by multidrug-resistant organisms and *Clostridioides difficile*. Other biomarkers, such as C-reactive protein and presepsin, are already being tested as guidance for shorter antimicrobial treatment, but more research is needed. Current evidence suggests that biomarkers, mainly procalcitonin, should be implemented in antimicrobial stewardship programs even in the COVID-19 era, when, although bacterial coinfection rate is low, antimicrobial overconsumption remains high.

## 1. Introduction

Early and appropriate antimicrobial treatment remains key for sepsis management [1]. It is, however, sometimes difficult even for the most experienced physician to rule-out an infection in the critically ill and withhold antibiotics. The appropriate duration of treatment for severe infections is also not fully described. Current sepsis guidelines recommend a shorter rather than longer duration of antimicrobial treatment, but the definite duration remains at the discretion of the treating physician [2]. Doubts and fear for relapse have led to injudicious broad-spectrum and unnecessarily long antimicrobial treatment adding up to the emergence of antimicrobial resistance. In 2019, about 5 million deaths have been associated with bacterial antimicrobial resistance, underlying the urgent need for tight infection control and robust antimicrobial stewardship programs [3].

A biomarker should be easily measured and interpreted as an indicator of biological or pharmacologic responses to a therapeutic intervention [4]. The optimal sepsis biomarker should be sensitive and specific enough to rule in/out diagnosis, predict unfavorable outcomes and evaluate the host’s response to treatment in order to encourage escalation or de-escalation; this is a strategy called “biomarker-guided treatment” [5]. More than one hundred biomarkers have been studied for sepsis management [6]. However, the only biomarker developed to guide antimicrobial treatment based on evidence coming from randomized clinical trials (RCTs) is procalcitonin (PCT). PCT is a precursor of the thyroid gland hormone calcitonin, and it is increased in the circulation during bacterial infection as a product of cells of mesenchymal origin. This review aims to present cumulative evidence from clinical trials, mainly RCTs, on the use of PCT-guidance in promoting antimicrobial stewardship for the critically-ill by restriction of injudicious antimicrobial treatment. Brief reference is also done to other biomarkers that are under consideration.

## 2. Results and Discussion

### 2.1. Antimicrobial Stewardship through PCT-Guidance for Lower Respiratory Tract Infections

PCT is the best studied biomarker to guide antimicrobial treatment in lower respiratory tract infections (LRTI). The majority of these RCTs shared a common design comparing an algorithm to start or discontinue antibiotics based on measurements of PCT, with standard-of-care (SOC); SOC was defined as start, continuation, or stop of antibiotics at the discretion of the treating physician and in accordance with local and international guidelines [7,8,9,10,11,12,13,14,15,16,17,18,19,20,21,22,23,24,25,26,27,28,29]. Available trials of PCT-guidance versus SOC are summarized in Table 1. Participants have a wide spectrum of symptoms, ranging from acute exacerbation of asthma and chronic obstructive pulmonary disease admitted in the Emergency Department, to severe community- or hospital-acquired pneumonia necessitating admission in medical wards or in the Intensive Care Unit (ICU). The common finding of all studies is the reduction of antimicrobial treatment duration with PCT-guidance. This reduction of antimicrobial treatment did not generate any safety signal as far as infection relapse, new infection, adverse events, or mortality are concerned. The ProHOSP trial studied the efficacy of PCT guidance directed to both start and stop of antibiotics. More precisely, 671 patients with LRTI received PCT-guided treatment and 688 received SOC [12]. For the PCT group, physicians were advised to start antibiotics when serum PCT was more than 0.25 ng/mL. Measurements were repeated on days 3, 5 and, 7 and stop of treatment was encouraged when levels decreased to more than 80% from the baseline. PCT-guidance led to a shorter antimicrobial treatment compared to SOC (5.7 versus 8.7 days, *p* < 0.05). However, the ProACT trial conducted in the USA a decade later, failed to show a similar effect. In the ProACT trial, mean duration of treatment for the 826 patients randomized in the PCT group was 4.2 days compared to that of 4.3 days for the 830 patients allocated in the SOC group (*p*: 0.87) [24]. One explanation for this lack of effect is the already reduced SOC duration of treatment in patients following local guidelines which does not allow any further benefit from the intervention to be shown. The majority of the first trials evaluating PCT-guidance provided such promising results that led to a switch in the current guidelines to a shorter duration of antimicrobial treatment for pneumonia [30] and to the approval of PCT guidance by the US Food and Drug Administration [31]. 

### 2.2. Antimicrobial Stewardship through PCT-Guidance in Sepsis

Critically ill and sepsis patients are the next most commonly studied population for benefit following PCT-guidance [32,33,34,35,36,37,38,39,40,41,42,43,44,45,46,47,48,49,50,51]. The efficacy of existing trials is summarized in Table 2. The majority of participants suffered from LRTI and intra-abdominal or urinary infections were less common. The majority of RCTs were conducted before Sepsis-3 implementation. One concern is that specific subgroups of patients, like pregnant and immunosuppressed, have been excluded from participation. Most of trials conclude that a PCT strategy reduces antimicrobial treatment duration without increase in adverse events and unfavorable outcomes. 

The PRORATA trial was the first large study evaluating PCT-guidance in ICU patients with suspicion of bacterial infection [36]. Three hundred and seven patients were randomized to PCT-guided treatment and 314 to SOC. For those in the PCT group, an algorithm of both initiation and cessation of antimicrobials was applied. When serum PCT was 0.5 ng/mL or more, physicians were encouraged to start antimicrobials and continue treatment until levels became less than 0.5 ng/mL in serial measurements or they decreased by at least 80% of the baseline value. The same algorithm was followed for every secondary infection episode until day 28 or discharge. The trial ended in a significant decrease in antimicrobial treatment duration from 14.3 days in SOC to 11.6 in PCT group (*p* < 0.0001). Mortality, relapse, and re-infection rate was similar between the two groups. There was however a trend for higher mortality with PCT-guidance after 60 days. 

The largest SAPS trial so far incorporated this knowledge and was designed to evaluate a stopping algorithm based on serial PCT measurements [44]. In the PCT group, physicians were encouraged to stop antimicrobials when PCT was less than 0.5 ng/mL on two consecutive days or PCT decreased by at least 80% of the baseline value. Mean antimicrobial duration was 5 days for 761 patients allocated to PCT group compared to 7 days for 785 patients allocated to SOC (*p* < 0.0001). Surprisingly, SAPS investigators came across a novel, interesting finding; PCT-guidance reduced both 28-day (19.6% vs. 25%, *p*: 0.0122) and 1-year mortality (34.8% vs. 40.9%, *p*: 0.0158). 

The recently published PROGRESS trial was the first trial conducted after the introduction of the Sepsis-3 definitions using the same stopping rule for antimicrobials as the SAPS trial [48]. PROGRESS was designed to provide an explanation of the findings of the SAPS trial on mortality. As a consequence, the primary endpoint of PROGRESS was the effect of long-term infectious complications in the critically ill, i.e., the incidence of new infection by multi-drug resistant organisms (MDRO) and *Clostridioides difficile* and mortality associated with baseline infection by MDRO or *C. difficile*. The incidence of these long-term complications after six months was 7.2% in the PCT and 15.3% in the SOC group (*p*: 0.045). Alongside this benefit, PCT guidance, decreased the length of antimicrobial treatment (5 vs. 7 days; *p* < 0.0001); and decreased 28-day mortality (15.2% vs. 28.2%; *p*: 0.02) among the 125 patients allocated in the PCT group compared to 131 patients allocated in the SOC group. The incidence of antibiotic-associated adverse events was strikingly decreased using PCT-guidance, in particular diarrhea and acute kidney injury (AKI); in the SOC arm, 36.6% of patients presented diarrhea and 17.6% AKI, compared to 19.2% (*p*: 0.002) and 7.2% (*p*: 0.01) in the PCT-guidance arm, respectively. Interestingly, the incidence of gut colonization by MDRO and *C. difficile* was similar between the two groups but the risk for clinical infection was significantly higher in colonized patients in the SOC but not in the PCT arm. These results indicate that long-term antibiotic exposure in the SOC arm could either affect the integrity of the mucosal barrier or modulate the composition of the gut microbiota resulting in the increased incidence of infections by MDRO and *C. difficile*.

Two trials similar in design to PROGRESS, are ongoing in France. The MultiCov trial (NCT04334850) is randomizing patients with severe COVID-19 into PCT-guided treatment or SOC. PCT-guidance is accompanied by sampling of respiratory secretions with multiplex PCR to identify bacterial pathogens [52]. The main aim of the study is to show a reduction in antibiotic exposure in the era of COVID-19 having as primary endpoint the number of antibiotic-free days until day 28 and among secondary outcomes the rate of colonization and/or infection by MDRO or *C. difficile* [53]. The MULTI-CAP trial randomizes patients with severe community-acquired pneumonia in the ICU to a combined PCT/multiplex respiratory PCR arm versus SOC; primary endpoint is antibiotic-free days until day 28.

The benefit disclosed by the larger ProHOSP and SAPS trial was further corroborated by smaller studies from developing countries [49,50,51] and meta-analyses [54,55,56,57,58,59,60,61,62]. A first meta-analysis was published in 2018 including a total of 4482 ICU patients and sub-analyzing patients meeting Sepsis-3 criteria [58]. PCT-guidance reduced 28-day mortality (OR 0.89; 95% CI: 0.80–0.99; *p*: 0.03) and mean duration of antibiotic treatment (−1.19 days; 95% CI: −1.73 to −0.66; *p* < 0.0001). Meta-analyses also confirmed reduction of antimicrobial treatment by PCT-guidance in special populations, such as patients with bacteremia [63], renal failure [64], or among the elderly [65]. Interestingly, some meta-analyses support that PCT-guidance is associated with decreased antimicrobial consumption and mortality only if cessation algorithms are applied [58]. A summary of published meta-analyses evaluating PCT-guidance is presented in Table 3 [54,55,56,57,58,59,60,61,62,63,64,65,66].

### 2.3. Real-World Data

Evidence supporting PCT-guidance for antimicrobial stewardship in critically ill patients, as already discussed, is from RCTs with different degrees of compliance to the PCT rule applied in each RCT, ranging from 44% up to 97%. It has not yet been clear if low adherence to PCT algorithms interferes with results and affects antimicrobial duration and mortality. Results of RCTs may not be in alignment with real-world data. Treating physicians participating in a RCT are influenced in decision making as they may feel under observation from the Sponsor or trial coordinators; this is namely the “Hawthorne effect” [67]. With this in mind, real-world evidence is mandatory. Soon after ProHOSP trial has been published, real world data supported compliance of physicians with the suggested algorithm as high as 72.5% [68].

Several implementation trials have investigated the effect of PCT-guidance in antimicrobial stewardship programs [69,70,71,72,73,74,75,76,77]. Main conclusions of these trials include (i) reduction in antimicrobial consumption; (ii) reduction in length of stay; (iii) reduction in hospitalization cost; and iv) no difference in infection-relapse of rehospitalization rate. Best implementation of the biomarker in real-world settings requires constant education of treating physicians for rightful use [78,79].

### 2.4. Antimicrobial Stewardship through Other Biomarkers

Other biomarkers have been also tested in antibiotic stewardship programs like serum C-reactive protein (CRP), serum presepsin, and interleukin (IL)-1β/IL-18 in bronchoalveolar lavage (BAL).

In a former trial, CRP was compared to PCT for the early stop of antibiotics. Discontinuation of antibiotics in the PCT arm was advised by more than 90% baseline decreases (n = 49) and in the CRP arm by more than 50% baseline decreases or values less than 25 mg/L (n = 45) [41]. Both strategies were non-inferior in terms of length of treatment, relapse rate, and ICU length of stay. A recent trial compared in a 1:1:1 randomization pattern, the clinical effectiveness of CRP-guided stop of antibiotics with fixed 7- and 14-day antibiotic durations in 504 hospitalized patients with gram-negative bacteremia [80]. Median antibiotic duration in the CRP group was seven days; clinical failure between the three arms of treatment was non-inferior. In another open-label RCT, CRP-guided antimicrobial treatment was compared to SOC in 130 ICU patients with sepsis and/or septic shock [81]. In the CRP arm, the biomarker was measured after five days from start of antibiotics and antibiotics were stopped when CRP decrease more than 50% or when it was found less than 35 mg/L. This strategy did reduce antibiotic duration or 28-day mortality.

Presepsin is the soluble form of CD14 (sCD14), an anchored glycoprotein expressed on monocytes and macrophages, serving as a receptor for bacterial lipopolysaccharide (LPS) [82]. Compared to CRP and PCT, presepsin appears advantageous in sepsis diagnosis, as it rises early, already in the first two hours after an infection. Recently, Xiao et al., conducted a prospective, multicenter, not randomized trial in China, comparing presepsin-guidance to SOC in sepsis [83]. In the presepsin group, physicians were advised to stop the antibiotics by serum concentrations lower than 350 pg/mL or any baseline decrease more than 80%. Antibiotic adjustment was encouraged when the blood presepsin concentration did not decline. Although the primary outcome (days without antibiotics at day 28) was achieved, mortality did not differ between treatment arms. 

In a recent trial conducted in the United Kingdom, 210 ICU patients with suspicion of ventilator-associated pneumonia (VAP) were allocated to a biomarker-guided approach (n: 104) or SOC (n: 106) [84]. In the biomarker-guided recommendation group measurements of IL-1β and IL-18 were performed in the bronchoalveolar lavage (BAL), and if concentrations were below a previously validated cutoff, clinicians were advised that VAP was unlikely and withheld antibiotics. The primary outcome was antibiotic-free days in the seven days following BAL; the trial did not achieve this endpoint. 

### 2.5. Antimicrobial Stewardship Using Biomarkers in the COVID-19 Era

In December 2019, a novel coronavirus, namely SARS-CoV-2, spread rapidly around the globe causing millions of cases of pneumonia leading to a rapid increase in hospitalizations and deaths. Patients presenting with COVID-19 pneumonia share common features with bacterial pneumonia (fever, cough, dyspnea, infiltrates in chest X-ray, and elevated inflammation markers) making the differential diagnosis troublesome. In severe cases, COVID-19 may resemble bacterial sepsis leading to multiorgan failure and requiring organ support in the ICU [85]. Although data are very heterogenous, unlike other viral respiratory diseases, bacterial co-infection at the time of hospital admission is rare in COVID-19; this may occur during hospital and/or ICU stay. A recent systematic review reports a rate of 8% of COVID-19 bacterial coinfection; surprisingly, the proportion of patients receiving antimicrobials is as high as 72% [86]. In such case, biomarkers, mainly PCT, may be useful in reducing unnecessary antimicrobial consumption.

A number of small case-series support that PCT is not elevated in COVID-19 patients, in contrast to other inflammation markers like CRP and ferritin [87,88,89]. The largest of these observational studies, conducted in New York, reports that only 16.9% of patients have PCT levels 0.5 ng/mL or more at hospital admission [90] and such high levels are associated with development of critical disease, admission in the ICU and increased risk for death [91,92,93]. A recent meta-analysis of 10 cohort studies including a total of 7716 patients estimated a pooled risk of 1.77 (95% CI, 1.38 to 2.29) for severe and critical COVID-19 by elevated PCT levels at admission, although results are highly heterogenous (I^2^:85.6%) [94]. Similarly, rise in PCT is associated with secondary bacterial infections, such as VAP and bacteremia [91,95,96,97]. PCT levels less than 0.25 ng/mL have been suggested as an optimal cut-off to rule out bacterial co-infection (negative predictive value 81%) and levels more than 1 ng/mL as optimal cutoff to rule in bacterial co-infection (positive predictive value 93%) [95].

It is questionable if pre-treatment with dexamethasone and tocilizumab in these patients is limiting the diagnostic performance of biomarkers. Kooistra et al. studied 190 ICU patients with COVID-19 having received different immunomodulatory agents and concluded that after treatment with dexamethasone and/or tocilizumab, CRP levels remain suppressed in case of a secondary bacterial infection but that the kinetics of PCT were not affected [98]. Thus, it is reasonable that CRP, which is elevated by the COVID-19-driven hyperinflammation and is suppressed by the immunomodulatory treatment does not represent the optimal biomarker to screen for bacterial complications in critically ill COVID-19. In contrast, PCT may inform about the early diagnosis of bacterial superinfection.

Real world data of PCT-guidance in COVID-19 support its use for a judicious antimicrobial approach. In a small retrospective cohort of 48 patients, median duration of antimicrobials was shorter if at least one PCT measurement was performed [99]. Similar results were also reported by Calderon et al. [100]. Williams et al. implemented a PCT guideline in the first 48 h after hospital admission of COVID-19 patients to withhold antibiotics with PCT less than 25 ng/mL [101]. Adherence to the guideline was high (77%). This strategy ended in lower defined daily doses (DDDs) per day alive, lower 28-day mortality, lower intubation, and ICU-admission rate. Staub et al. reported an increase in the antimicrobial usage during the COVID-19 pandemic compared with the pre-COVID era, but this usage decreased again after implementation of a guidance team using biomarkers [102]. A summary of PCT trials in COVID-19 patients is provided in Table 4.

In contrast with a plethora of RCTs evaluating PCT-guidance in sepsis, such high-quality data are missing for COVID-19. MultiCov is an ongoing RCT in France, evaluating PCT-guided treatment in combination with FilmArray syndromic diagnostics compared to SOC to prove a benefit in the number of antibiotic-free days, mortality, rate of bacterial superinfection and rate of colonization/infection by MDRO and/or *C. difficile* (NCT04334850) [52]. Results of the trial will be of great interest to guide appropriate antimicrobial administration in the COVID-19 era.

## 3. Materials and Methods

To address the aim of this review and to present recent evidence in biomarker-guidance in the critically ill with emphasis on antimicrobial stewardship, the authors searched independently “Pubmed” and the database “clinicaltrials.gov” under the terms: “sepsis”, “COVID-19”, “infection”, “critically ill”, “intensive care unit”, “biomarker guidance”, “guided treatment”, “procalcitonin”, and “c-reactive protein” about randomized clinical trials and observational studies conducted in humans aged equal to or older than 18 years old, published in English, with emphasis on trials published in the last decade (2012–2022). The literature search yielded 11,791 records; after removal of duplicates and records with irrelevant titles, 611 were screened in full-text by the reviewers. After applying exclusion criteria, 102 studies were finally analyzed (Figure 1).

## 4. Conclusions

Biomarkers, mainly procalcitonin, may guide antimicrobial treatment with safety in two directions; (i) improve patient outcomes by reduction in antibiotic-associated adverse events and (ii) globally reduce the high burden of antimicrobial resistance. Procalcitonin-guidance of antimicrobial treatment for the critically ill decreases the length of antimicrobial treatment, the length of stay (Hospital/ICU), and the cost of hospitalization and in parallel, the strategy improves both short- and long-term outcomes including mortality and rate of secondary infections by MDRO and *C. difficile*. In the COVID-19 era, data suggest a crucial role of the biomarker to reduce unnecessary antimicrobial overuse. Thus, biomarkers should be incorporated in antimicrobial stewardship programs and physicians’ education is key for their appropriate application in every day clinical practice.

## Figures and Tables

**Figure 1 antibiotics-11-00367-f001:**
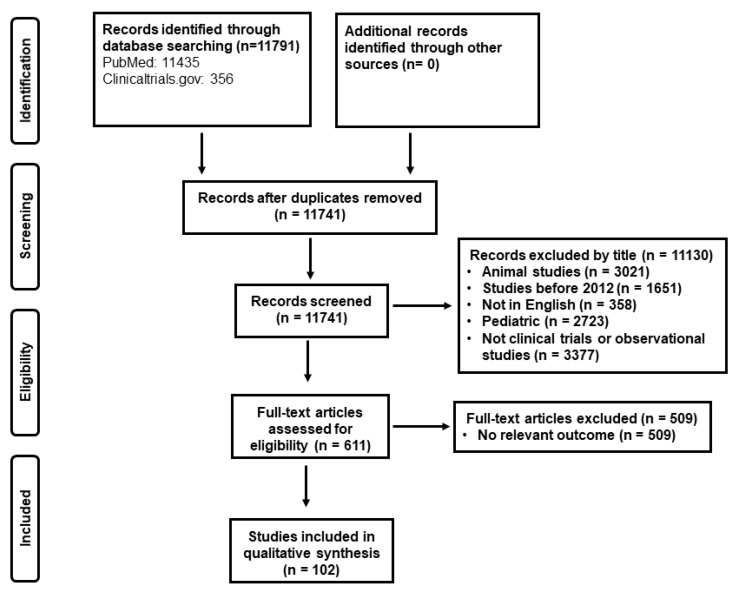
Study selection.

**Table 1 antibiotics-11-00367-t001:** Summary of randomized trials evaluating Procalcitonin (PCT)-guided antimicrobial treatment in patients with infections outside the Intensive Care Unit (ICU).

Ref	Trial Setting	PCT Algorithm Applied	N of Patients	Main Results
[7]	LRTI—ED Single-center—Switzerland	Initiation-cessation	PCT: 124 SOC: 119	Prescription of antimicrobials: 44% vs. 83%, *p* < 0.0001 LOT: 10.3 vs. 12.8 days, *p* < 0.0001 Decreased cost
[8]	CAP (requiring hospitalization) Single-center—Switzerland	Initiation-cessation	PCT: 151 SOC: 151	Prescription of antimicrobials: 85% vs. 99%, *p* < 0.0001 LOT: 5.8 vs. 12.9 days, *p* < 0.0001 Decreased cost
[9]	COPD exacerbation—ED Single-center—Switzerland	Initiation-cessation	PCT: 113 SOC: 113	Prescription of antimicrobials: 40% vs. 72%, *p* < 0.0001
[10]	Symptoms compatible with respiratory (upper/lower) infection—prehospital Multicenter—Switzerland	Initiation-cessation	PCT: 230 SOC: 223	Restriction in activity at day 14: 0.14 (95% CI: −0.53 to 0.81) Prescription of antimicrobials: decrease 72% (95% CI: 66–78)
[11]	CAP (requiring hospitalization) Multicenter—Denmark	Initiation with PCT > 0.25 ng/mL	PCT: 103 SOC: 107	LOT: 5.1 vs. 6.8 days, *p* = 0.007
[12]	CAP (requiring hospitalization) Multicenter—Switzerland	Initiation-cessation	PCT: 687 SOC: 694	Total adverse outcomes: 15.4% vs. 18.9%, OR −3.5 (95% CI: −7.6 to 0.4) LOT: 5.7 vs. 8.7 days, *p* < 0.05 AE due to antimicrobials: 19.8% vs. 28.1%, *p* < 0.05
[13]	Symptoms compatible with respiratory (upper/lower) infection—prehospital	Initiation-cessation	PCT: 275 SOC: 275	Restriction in activity at day 14: 0.04 (95% CI: 0.73 to 0.81) Prescription of antimicrobials: 21.5% vs. 36.7%, *p* < 0.0005
[14]	CAP (requiring hospitalization) Single-center—Shanghai	Initiation-cessation	PCT: 81 SOC: 81	Prescription of antimicrobials: 84.4% vs. 97.5%, *p* = 0.004 LOT: 5 vs. 7 days, *p* < 0.001
[15]	Acute asthma exacerbation Single-center—Shanghai	Initiation	PCT: 132 SOC: 133	Prescription of antimicrobials: 46.1% vs. 74.8%, *p* < 0.01
[16]	Aspiration pneumonia Single-center—Japan	If initial PCT < 0.5 ng/mL treat 3 days; if 0.5–1.0 treat for 5 days; if >1.0 treat for 7 days; stop with decrease ≥90%	PCT: 53 SOC: 52	Relapse (30 days): 25% vs. 37.5%, *p* = 0.19 LOT: 5 vs. 8 days, *p*< 0.0001
[17]	Acute asthma exacerbation Single-center—Shanghai	Initiation	PCT: 90 SOC: 90	Prescription of antimicrobials: 48.9% vs. 87.8%, *p*< 0.001 LOT: 6 vs. 6 days, *p* = 0.198 Exacerbation (1 year): 78.8% vs. 82.1%, *p* = 0.586
[18]	COPD exacerbation Multicenter—Italy	Stop after 3 days if PCT < 0.25 ng/mL; if not treat for 10 days	PCT: 88 SOC: 90	Exacerbation rate difference (6 months): 4.04% (90% CI: −7.23 to 15.31)
[19]	LRTI (requiring hospitalization)—ED Single-center—USA	Initiation, combined with multiplex PCR	PCT:151 SOC: 149	LOT: 3 vs. 4 days, *p* = 0.42 Duration of symptoms: 16 vs. 20 days, *p* = 0.03
[20]	COPD exacerbation Single-center—Denmark	Initiation-cessation	PCT: 62 SOC: 58	LOT: 3.5 vs. 8.5 days, *p* = 0.0169 Patients (%) under treatment ≥5 days: 41.9 vs. 67.2, *p* = 0.006
[21]	After stroke Multicenter—International	Initiation	PCT: 112 SOC: 115	modified Rankin Scale (3 months): 4 vs. 4, *p* = 0.452 Prescription of antimicrobials: 63% vs. 45%, *p* = 0.01
[22] *	COPD exacerbation Single-center—USA	Initiation	Before:139 After: 166	LOT: 3 vs. 5.3 days, *p* = 0.01 Length of hospital stay: 2.9 vs. 4.1 days, *p* = 0.01 Rehospitalization (30 days): 16.6% vs. 14.5%, *p* = 0.25
[23]	COPD exacerbation Multicenter—France	Initiation-cessation	PCT: 151 SOC: 151	Mortality (3 months): 20% vs. 14%, LOT: no difference
[24]	LRTI—ED Multicenter—USA	Initiation-cessation	PCT: 826 SOC: 830	LOT: 4.2 vs. 4.3 days, difference −0.05 (95% CI −0.6 to 0.5) Prescription of antimicrobials (30 days): 57% vs. 61.8% Length of hospital stay: 4.7 vs. 5.0 days
[25]	Fever ≥ 38.2 °C—ED (main infection [40%] respiratory) Two-center—Netherlands	Initiation	PCT: 275 SOC: 276	Prescription of antimicrobials: 73% vs. 77%, *p* = 0.28 Readmission at ED (14 days): 7% vs. 10%, *p* = 0.20 Hospitalization: 74% vs. 81%, *p* = 0.10 Mortality (30 days): 2% vs. 4%, *p* = 0.11
[26] **	LRTI (requiring hospitalization)—ED Single-center—USA	Initiation-cessation	After: 174 Before: 200	LOT: 5 vs. 6 days, *p* = 0.052 LOT-pneumonia: 6 vs. 7 days, *p* = 0.045 LOT-COPD exacerbation: 3 vs. 4 days, *p* = 0.01
[27]	CAP—ED Multicenter—France	Initiation-cessation	PCT: 142 SOC: 143	LOT:10 vs. 9 days, *p* = 0.21 AE: 15% vs. 20%, difference 5% (95% CI: −4 to 14%) Mortality (30 days): 1% vs. 2%, *p* > 0.05
[28] ***	CAP and/or HCAP Single-center—Japan	Cessation cutoff 0.2 ng/mL	PCT: 116 SOC: 116	LOT: 8 vs. 11 days, *p* < 0.001 Relapse (30 days): 4.3% vs. 6.0%, *p* = 0.5541
[29]	Symptoms of acute heart failure—ED Multicenter—International	Initiation cutoff 0.2 ng/mL	PCT: 370 SOC: 372	Mortality (90 days): 10.3% vs. 8.2%, *p* = 0.316 Mortality (30 days): 6.8% vs. 4.3%, *p* = 0.152 Prescription of antimicrobials: 18% vs. 14%, *p* = 0.145 Rehospitalization (30 days): 17.3 vs. 9.7%, *p* = 0.004

* retrospective before-after study; ** prospective before-after study; *** patient-historical control study. Abbreviations: CAP—community-acquired pneumonia; COPD—chronic obstructive pulmonary disease; CI—confidence interval; ED—emergency department; HCAP—healthcare-associated pneumonia; ICU—intensive care unit; LOT—length of therapy; LRTI—lower respiratory tract infection; PCT—procalcitonin; SOC—standard-of-care.

**Table 2 antibiotics-11-00367-t002:** Summary of randomized trials evaluating Procalcitonin (PCT)-guided antimicrobial treatment in critically ill patients with severe infection/sepsis in the Intensive Care Unit (ICU).

Ref	Trial Setting	PCT Algorithm Applied	N of Patients	Main Results
[32]	Severe sepsis and septic shock (65% respiratory infections) Single-center—Switzerland	Cessation if ≥90% decrease or PCT < 0.25 ng/mL	PCT: 31 SOC: 37	LOT: 3.5 vs. 6 days, *p* = 0.15 (ITT) 6 vs. 10 days, *p* = 0.003 (PP) Length of ICU stay: 4 vs. 7 days, *p* = 0.02
[33]	Severe sepsis after intraabdominal surgery Single-center—Germany	Cessation if PCT < 1 ng/mL for 3 consecutive days	PCT: 14 SOC: 13	LOT: 6.6 vs. 8.3 days, *p* < 0.001
[34]	Sepsis Single-center—Germany	Cessation if PCT < 1 ng/mL or ≥65% decrease for 3 serial days	PCT: 57 SOC: 53	LOT: 5.9 vs. 7.9 days, *p* < 0.001 Length of ICU stay: 15.5 vs. 17.7 days, *p* = 0.046
[35]	VAP Multicenter—Switzerland and USA	Initiation-cessation	PCT: 50 SOC: 51	LOT: 7 vs. 11 days, *p* = 0.044
[36]	Sepsis (mainly [70%] respiratory infections) Multicenter—France	Initiation-cessation	PCT: 307 SOC: 314	LOT: 6.1 vs. 9.9 days, *p* < 0.0001 Relapse: absolute difference 1.4% Reinfection: absolute difference 3.6%
[37]	Suspected infection Multicenter—Denmark	Up-escalation when PCT > 1.0 ng/mL	PCT: 604 SOC: 596	Significantly higher antimicrobial consumption in PCT group
[38]	Suspected infection (60% respiratory infections) Single-center—Belgium	Initiation	PCT: 258 SOC: 251	Antimicrobial consumption (% days in ICU): 62.6 vs. 57.7, *p* = 0.11
[39]	Acute pancreatitis Single-center—China	Initiation-cessation PCT cutoff: 0.5 ng/mL	PCT: 35 SOC: 36	LOT: 10.89 vs. 16.06 days, *p* < 0.001 Length of stay: 16.66 vs. 23.81 days, *p* < 0.001
[40]	Sepsis Single-center—Brazil	Cessation if PCT < 0.5 ng/mL or ≥90% decrease	PCT: 42 SOC: 39	LOT: 10 vs. 11 days, *p* = 0.44 (ITT) 9 vs. 13 days, *p* = 0.008 (PP)
[41]	Sepsis (60% respiratory infections) Two-center—Brazil	Cessation PCT < 0.1 ng/mL or ≥90% from baseline CRP < 25 mg/L or ≥50% decrease from baseline	PCT: 50 CRP: 47	LOT: 7 vs. 6 days, *p* = 0.06 Mortality: 32.7% vs. 33.3%, *p* = 1.000
[42]	Sepsis Multicenter—France	Initiation-cessation	PCT: 27 SOC: 26	Patients (%) under treatment at day 5: 67 vs. 81, *p* = 0.24
[43]	Suspected sepsis Multicenter—Australia	Initiation-cessation Cessation when PCT < 0.10 ng/mL or ≥90% decrease from baseline	PCT: 196 SOC: 198	LOT: 9 vs. 11 days, *p* = 0.58 Total doses of antimicrobials: 1200 vs. 1500, *p* = 0.001
[44]	Sepsis Multicenter—Netherlands	Cessation if PCT < 0.5 ng/mL or ≥80% from baseline for 2 serial days	PCT: 761 SOC: 785	LOT: 5 vs. 7 days, *p* < 0.0001 Mortality (28 days):19.6% vs. 25%, *p* = 0.0122 Mortality (1 year): 34.8% vs. 40.9%, *p* = 0.0158
[45]	Sepsis Multicenter—Germany	Cessation if PCT < 1.0 ng/mL or ≥50% decrease	PCT: 552 SOC: 537	Mortality: 25.6% vs. 28.2%, *p* = 0.34 Antimicrobials/1000 ICU days: 823 vs. 862, decrease 4.5%, *p* = 0.02
[46]	Severe sepsis and/or septic shock Multicenter—Korea	Cessation if PCT < 0.5 ng/mL or ≥90% from baseline	PCT: 23 SOC: 29	LOT:10 vs. 13 days, *p* = 0.078 (ΙΤΤ), 8 vs. 14 days, *p* < 0.001 (PP) Mortality (28 days): 17% vs. 21%, *p* = 0.709
[47] *	VAP Multicenter—France	Initiation-cessation	PCT: 76 No-PCT: 81	LOT: 8 vs. 9.5 days, *p* = 0.02 Death and/or relapse: 51.3% vs. 46.9%, *p* = 0.47
[48]	Sepsis-3 Multicenter—Greece	Cessation if PCT < 0.5 ng/mL or ≥80% decrease from baseline	PCT: 125 SOC: 131	LOT: 5 vs. 10 days; *p* < 0.001 Mortality (28 days): 15.2% vs. 28.2%, *p* = 0.02
[49] **	Surgical trauma Single center—South Africa	Cessation if PCT < 0.5 ng/mL or ≥80% from baseline	PCT: 40 SOC: 40	LOT: 9.3 vs. 10.9 days, *p* = 0.10 Mortality: 15% vs. 30%, *p* = 0.045
[50]	VAP Single center—Malaysia	Cessation if PCT < 0.5 ng/mL or ≥80% from baseline	PCT: 43 SOC: 42	LOT: 10.28 vs. 11.52 days, difference −1.25 (95%CI −2.48 to 0.01), *p* = 0.049
[51]	Sepsis and septic shock Single center—India	Cessation if PCT < 0.01 ng/mL or ≥80% from baseline	PCT: 45 SOC: 45	LOT: 4.98 vs. 7.73 days, *p* < 0.001 Length of ICU stay: 5.98 vs. 8.80 days, *p* < 0.001 Secondary infections: 4.4% vs. 26.7%, *p* = 0.014 Mortality: 8.9% vs. 15.6%, *p* = 0.522 Readmission: no difference

* prospective observational trial; ** prospective two-period cross-over trial. Abbreviations: CI—confidence interval; ICU—intensive care unit; ITT—intention to treat; LOT—length of therapy; PCT—procalcitonin; PP—per protocol; SOC—standard-of-care; VAP—ventilator-associated pneumonia.

**Table 3 antibiotics-11-00367-t003:** Summary of meta-analyses evaluating Procalcitonin (PCT)-guided antimicrobial treatment in critically ill patients.

Ref	N of Trials	N of Patients	Focus of Interest	Main Results
[54]	10	1215	NA	Antibiotic duration (days): −1.28 days (95% CI −1.95 to −0.61) Mortality: RR 0.81 (95% CI 0.65 to 1.01)
[55]	13	5136	Antibiotic Initiation, Cessation, or Mixed Strategies	Antibiotic duration (days): −1.66 (95% CI −2.36 to −0.96) Mortality: RR 0.87 (95% CI 0.76 to 0.98)
[56]	26	6708	Acute respiratory infections	Antibiotic duration (days): −2.4 (95% CI −2.71 to −2.15) Mortality: OR 0.83 (95% CI 0.70 to 0.99) Antibiotic-related side-effects: OR 0.68 (95% CI 0.57 to 0.82)
[57]	11	4482	Subgroup of sepsis-3	Antibiotic duration (days): −1.19 (95% CI −1.73 to −0.66) Mortality: OR 0.89 (95% CI 0.80 to 0.99) Sepsis-3, OR 0.86 (95% CI 0.76 to 0.98)
[58]	15		Antibiotic Initiation, Cessation, or Mixed Strategies	Antibiotic duration (days): −1.26 (*p* < 0.001) and −3.10 (*p* = 0.04) for cessation and mixed strategies, respectively Mortality: OR 1.00 (95% CI 0.86 to 1.15), 0.87 (95% CI 0.77 to 0.98), and 1.01 (95% CI 0.80 to 1.29) for the initiation, cessation, and mixed procalcitonin strategies, respectively
[59]	10	3489	Suspected or confirmed sepsis	Antibiotic duration (days): −1.49 (95% CI −2.27 to −0.71) Mortality: RR 0.90 (95% CI 0.79 to1.03)
[60]	16	5158	Subgroup (5 trials) with high algorithm adherence	Mortality: RR 0.89 (95% CI 0.83 to 0.97) In high algorithm adherence, RR 0.93 (95% CI 0.71 to 1.22)
[61]	16	6452	NA	Antibiotic duration (days): −0.99 (95% CI −1.85 to −0.13), *p* = 0.02 Mortality: OR 0.90 (95% CI 0.80 to1.01)
[62]	14	4744	NA	Antibiotic duration (days): −1.23 (95% CI −1.61 to −0.85) Mortality: OR 0.91 (95% CI 0.82 to 1.01)
[63]	13	523 (IPD)	Positive blood culture	Antibiotic duration (days): −2.86 (95% CI −4.88 to −0.84) Mortality: 16.6% vs. 20.0%, *p* = 0.263
[64]	15	5002 (IPD)	Kidney function (3 groups: GFR > 90, GFR 15–89 and GFR < 15)	Antibiotic duration (days): −2.06 (95% CI −2.87 to −1.25), −1.72 (95% CI −2.29 to −1.16), −2.49 (95% CI −3.59 to −1.40), *p*_interaction_ = 0.336. Overall, −2.01 (95% CI −2.45 to −1.58) Mortality: OR 1.08 (95% CI 0.79 to 1.49), 0.74 (95% CI 0.63 to 0.87), 1.03 (95% CI 0.83 to 1.29), *p*_interaction_ = 0.888. Overall, 0.88 (95% CI 0.78 to 0.98)
[65]	28	9421 (IPD)	Age (4 groups: <75, 75–80, 81–85 and >85 years)	Antibiotic duration per age group (days): Less than 75 years: −1.99 (95% CI −2.36 to −1.62); 75–80 years: −1.98 (95% CI −2.94 to −1.02); 81–85 years: −2.20 (95% CI −3.15 to −1.25), more than 85 years: −2.10 (95% CI −3.29 to −0.91), *p*_interaction_ = 0.654. Overall, −2.01 (95% CI −2.32 to −1.69) Mortality: Less than 75 years: OR 0.87 (95% CI 0.76 to 1.00); 75–80 years 0.86 (95% CI 0.67 to 1.10); 81–85 years 1.19 (95% CI 0.76 to 1.06), *p*_interaction_ = 0.891. Overall, 0.90 (95% CI 0.81 to 1.00)
[66]	12	42,921	NA	Antibiotic duration (days): 1.98 days (95% CI: −2.76, −1.21) Mortality: RR 0.89 (95% CI 0.79 to 0.99) ICU-length of stay (days): −1.21 (95% CI −4.16 to 1.74)

Abbreviations: CI—confidence interval; GFR—glomerular filtration rate; IPD—individual patient data; NA—non applicable; OR—odds ratio; PCT—procalcitonin; RR—risk ratio.

**Table 4 antibiotics-11-00367-t004:** Summary of trials evaluating Procalcitonin (PCT) in COVID-19 patients.

Ref	Type and Setting of Study	N of Patients	Severity of COVID-19	Main Results
[87]	Observational February-March 2020 Single-center, USA	21	Critical ICU patients	Median PCT 1.8 (0.12–9.56)
[88]	January-February 2020 Single-center, Wuhan China	138	Hospitalized Both critical/non-critical	PCT ≥ 0.05 ng/mL in 35.5% of patients Higher levels in patients requiring ICU
[90]	Retrospective case series March 2020 Two-center, USA	393	Hospitalized Both critical/non-critical	PCT ≥ 0.05 ng/mL in 16.9% of patients Higher levels in patients requiring intubation
[91]	Retrospective observational March-April 2020 Single-center, USA	324	Hospitalized Both critical/non-critical	PCT for prediction of bacteremia, AUC 0.81 (0.64–0.98) PCT for prediction of bacterial pneumonia, AUC 0.75 (0.64–0.86)
[92]	Retrospective observational March-April 2020 Multicenter, UK	224	Hospitalized Both critical/non-critical	PCT > 0.5 ng/mL in 16.5% of patients PCT associated with increased risk of death (*p* = 0.0004)
[93]	Retrospective observational March-June 2020 Multicenter, Spain	777	Critical ICU patients	PCT 0.64 (0.17–1.44) ng/mL in non-survivors compared to 0.23 (0.11–0.60) ng/mi in survivors, *p* < 0.01
[95]	Observational Single-center, Netherlands	66	Critical ICU patients	PCT > 1.00 ng/mL at admission rule in secondary bacterial infection PCT < 0.25 ng/mL at admission rule out secondary bacterial infection
[96]	Retrospective observational March-June 2020 Single-center, UK	65	Critical ICU patients	PCT rise in 81.5% of patients PCT rise in 97% of patients with confirmed VAT/VAP and/or BSI
[97]	Retrospective observational March-October 2020 Single-center, Germany	99	Hospitalized Both critical/non-critical	PCT of patients with secondary bacterial infection 0.4 ng/mL versus 0.1 of those without, *p* = 0.016 cut-off 0.55 ng/mL: sensitivity 91%, specificity 81% for bacterial infection

Abbreviations: AUC—area under the curve; BSI—bloodstream infection; ICU—intensive care unit; PCT—procalcitonin; VAT—ventilator-associated tracheobronchitis; VAP—ventilator-associated pneumonia.

## Data Availability

All data are presented throughout the manuscript.

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
