# Peer review of "Antimicrobial Stewardship Using Biomarkers: Accumulating Evidence for the Critically Ill"

_antibiotics, 2022, doi:10.3390/antibiotics11030367_

Round 1
Reviewer 1 Report
Dear Authors:
The manuscript by Kyriazopoulou et al has demonstrated that biomarkers should be incorporated in antimicrobial stewardship programs and physicians’ education is key for their appropriate application in every day clinical practice.
1. The manuscript needs linguistic improvement.
2. Could you clarify the criteria of selection of literature, better to show it in a flow chart in manuscript please.
Best,
Author Response
The manuscript by Kyriazopoulou et al has demonstrated that biomarkers should be incorporated in antimicrobial stewardship programs and physicians’ education is key for their appropriate application in every day clinical practice. The manuscript needs linguistic improvement.
Reply: Every effort was made to improve linguistically the manuscript.
Could you clarify the criteria of selection of literature, better to show it in a flow chart in manuscript please.
Reply: This is now clarified in the Section “Materials and Methods” and a new figure, namely Figure 1, has been added.
Reviewer 2 Report
The submitted manuscript is well written, informative and useful for daily practice and future research.
I would suggest to the authors to present in Materials and Methods more information about literature research. How many studies were retrieved using the terms authors used, how many of them were rejected etc.
Author Response
The submitted manuscript is well written, informative and useful for daily practice and future research.
Reply: We thank the reviewer for the positive evaluation of our manuscript.
I would suggest to the authors to present in Materials and Methods more information about literature research. How many studies were retrieved using the terms authors used, how many of them were rejected etc.
Reply: This is now clarified in the Section “Materials and Methods” and a new figure, namely Figure 1, has been added.
Reviewer 3 Report
Antimicrobial Stewardship Using Biomarkers: Accumulating Evidence for the Critically Ill
Technical Comments to the Author
The work is well structured, well written and easy to follow. It also addresses a topic that creates great interest in the scientific community, and it focuses on the current advances in the management of critically ill patients, using biomarkers as a guide for antimicrobial treatment. The subject is perfectly in line with the "Antibiotics" journal.
The review respects writing guide of journal.
Remarks to the Author
I suggest minor comments.
Minor comments
1) Implement the paragraph on materials and methods (3) with the precise number of studies used for writing the review and those eliminated. Added an image of a flow diagram, showing the process of study selection.
2) The “Conclusions” are very brief. The importance of the procalcitonin biomarker is not emphasized. I suggest expanding this paragraph, synthesizing the key points of the review.
3) English language and style are fine/minor spell check required
Author Response
The work is well structured, well written and easy to follow. It also addresses a topic that creates great interest in the scientific community, and it focuses on the current advances in the management of critically ill patients, using biomarkers as a guide for antimicrobial treatment. The subject is perfectly in line with the "Antibiotics" journal. The review respects writing guide of journal.
Reply: We thank the reviewer for the positive evaluation of our manuscript.
I suggest minor comments.
1) Implement the paragraph on materials and methods (3) with the precise number of studies used for writing the review and those eliminated. Added an image of a flow diagram, showing the process of study selection.
Reply: This is now clarified in the Section “Materials and Methods” and a new figure, namely Figure 1, has been added.
2) The “Conclusions” are very brief. The importance of the procalcitonin biomarker is not emphasized. I suggest expanding this paragraph, synthesizing the key points of the review.
Reply: This has been expanded in lines 313-318 of the revised manuscript.
3) English language and style are fine/minor spell check required
Reply: Every effort was made to improve linguistically the manuscript.
Round 2
Reviewer 1 Report
Strongly suggest for publishing